# Content Analysis of #Postpartumbody Images Posted to Instagram

**DOI:** 10.3390/healthcare10091802

**Published:** 2022-09-19

**Authors:** Megan L. Gow, Hiba Jebeile, Natalie B. Lister, Heike Roth, Helen Skouteris, Heidi Bergmeier

**Affiliations:** 1Children’s Hospital Westmead Clinical School, The University of Sydney, Sydney, NSW 2145, Australia; 2Discipline of Paediatrics, School of Clinical Medicine, The University of New South Wales, Sydney, NSW 2031, Australia; 3The Institute of Endocrinology and Diabetes, The Children’s Hospital at Westmead, Sydney, NSW 2145, Australia; 4Faculty of Health, University of Technology Sydney, Sydney, NSW 2007, Australia; 5Health and Social Care Unit, School of Public Health and Preventive Medicine, Monash University, Melbourne, VIC 3004, Australia

**Keywords:** postpartum, body image, social media, women, Instagram, health information

## Abstract

Understanding the imagery on social media targeting postpartum women is an important step in determining the utility of Instagram as a potential avenue for targeting public health messages to this group. This study (1) describes the content of images on Instagram tagged with #postpartumbody and; (2) compares images from ‘Top’ posts (‘trending’ or ‘popular’) with ‘Recent’ posts. 600 images tagged with #postpartumbody (300 ‘Top’ and 300 ‘Recent’) were systematically captured from Instagram and coded using a predefined framework. Images of women were coded for adiposity, muscularity, pose and attire. Chi-square tests were used to compare ‘Top’ and ‘Recent’ posts. Most (n = 409) images were of a woman who generally had low/average adiposity (91%) and little-to-none/some visible muscle definition (93%). Most women (52%) were posing in a non-specific manner, 5% were posing to accentuate a postpartum body feature and 40% were wearing fitness attire. Compared with ‘Recent’, ‘Top’ posts were less likely to be text-focused (*p* < 0.001), photos of food (*p* < 0.001) or linked to a product/program (*p* < 0.001). Women of lower adiposity are more likely to post images of themselves on Instagram tagged with #postpartumbody than women of higher adiposity, which may reflect increased body pride in this group, but could reduce body satisfaction for some viewers. Conveying health information on Instagram may be necessary to interrupt potentially harmful content.

## 1. Introduction

Optimising both physiological and psychological health postnatally is paramount to the future health of the child as well as the mother [1]. However, targeting health information to women in the postpartum period can be challenging as many aspects of life are undergoing change including disruptions to sleep patterns and the logistics of caring for a newborn baby. Despite these challenges, the postpartum period is a time when women are more health conscious and open to lifestyle behaviour change for a variety of reasons [2]. Therefore, provision of health information via platforms that are both flexible and easily accessible may be particularly important for women during this life stage.

Globally, 39.0% of Instagram’s billion monthly active users are females aged 18 to 44 years [3]. With 63% of users logging in at least daily, Instagram presents a possible avenue to target easily accessible health messages for women in the postpartum period. However, Instagram use has been attributed to reduced body satisfaction, particularly in young women [4,5,6]. This may be due to more self-objectification and body surveillance secondary to women comparing their own body to the bodies of other women observed on social media [7]. Furthermore, images on Instagram of real people may indeed have been manipulated prior to being uploaded to present an ‘ideal’ rather than ‘real’ version of themselves [8]. Frequent viewing of such images has been demonstrated to lead to body dissatisfaction and mood disturbances [9]. This is particularly important for postpartum women, who are already a vulnerable group at increased risk of body dissatisfaction [10]. Objectification theory provides a framework for understanding this internalization of objectification of the female body in the media, including social media platforms such as Instagram [11].

To ensure safety of intervention delivery, it is essential that careful planning take place before health initiatives are conducted via social media platforms [12,13]. Therefore, understanding the content of images on social media that target women in the postpartum period is an important first step in determining whether social media is an appropriate platform for conveying health information to postpartum women. 

The primary aim of this study was to describe the content of images uploaded to Instagram that women in the postpartum period are likely to be exposed to. The secondary aim was to describe and compare the content of the most ‘Recent’ images with ‘Top’ images, which are those trending/popular posts. In line with objectification theory and other research describing the content of Instagram images, it was hypothesized that the majority of women in images would depict a thin and/or muscular body shape.

## 2. Materials and Methods

This content analysis involved analysing posts tagged with #postpartumbody on Instagram at a random timeslot: Sunday afternoon/Monday morning (Australian Eastern Standard Time) in October 2020. Hashtags (#) are used to label an image with a particular word or phrase. Extensive searching of hashtags on Instagram identified #postpartumbody, compared with other hashtags, to be a highly utilised hashtag with specific relevance to the postpartum period. When the search was conducted in October 2020 there were 1.3 million images ever uploaded to Instagram tagged with #postpartumbody. At time of manuscript submission in September 2022, this figure had increased to 2.0 million which is equivalent to approximately 1000 images uploaded to Instagram tagged with #postpartumbody daily during this time period. Examples of other relevant but less utilised hashtags considered included #postpartumweightloss (565,000 posts in October 2020), #postpartumdiet (23,000 posts in October 2020), #postpartumnutrition (30,000 posts in October 2020), #postbabybody (300,000 posts in October 2020), #afterbabybody (140,000 posts in October 2020), #mumbod (100,000 posts in October 2020), #mombod (1.3 million posts in October 2020), #postpartumfitness (1.1 million posts in October 2020), #postpartumhealth (156,000 posts in October 2020), #mumtum (59,700 posts in October 2020), and #healthymum (374,000 posts in October 2020). 

Given that this is an emerging area of research with evolving methodology a sample of 600 images was captured for coding and analysis. This was based on methodology used by previous studies analysing body image and diet related imagery on Instagram, including: three unique studies which each captured 600 images posted to #weightloss [14], #cheatmeal [15] and #fitspiration [16]; another study captured, coded and analysed 300 images in total from two different hashtags (150 images each hashtag) [17]; and another which captured 200 images each from two different hashtags (400 images in total) [18]. The 600 images were systematically captured using an Instagram profile of an Australian female of average childbearing age (31.2 years) [19], created and used only for the purpose of this study. Of the 600 images, 300 were captured from the ‘Recent’ posts category, indicating images most recently uploaded to Instagram by users tagged with #postpartumbody and 300 from the ‘Top’ posts category, indicating posts that are ‘trending’ or ‘popular’ with highest recent engagement [20]. 

All 600 captured images were included and coded as part of this content analysis, including those appearing more than once and/or in both the ‘Top’ and ‘Recent’ posts category, to ensure the data would serve as a true representation of the images the user would view when searching #postpartumbody. Data on the profile holder, or text and comments associated with the post were not collected. When multiple images were uploaded as part of the same post, the first image was selected for coding. Similarly, when a video was uploaded, the image visible at the start of the video was captured for coding. 

Ethical approval was not obtained for this study as it included the collation and analysis of data available in the public domain.

### 2.1. Coding Framework

Each of the 600 images were coded using the codebook outlined in Table 1 which was adapted from a previous study [14] to include categories that focused on content relevant to the study aims and user profile (i.e., female 31.2 years) and were also informed by objectification theory [11]. First, the overall image category was determined. Images of food, a non-pregnant woman, a woman and baby(ies)/child(ren), before-and-after photos and text-focused images (including cartoons, quotes and informative images) were coded further based on previously defined categories (Table 1). Text-focused images were coded based on the information presented, not the accuracy of the information. Images linked to a product or program were identified if a logo, Instagram account, hashtag or product label was visible in the image.

### 2.2. Coding Procedure and Inter-Coder Reliability

Two authors completed coder training using an iterative process of consensus coding for the first 50 images. Coders then independently and in duplicate completed coding for 20% of images (n = 120; n = 60 ‘Recent’ images and n = 60 ‘Top’ images) to determine the inter-coder reliability using percent agreement and Cohen’s kappa [21]. Across all variables there was a mean agreement of 83% and Cohen’s kappa range 0.333 to 1.000 (Table 1) with poorer agreement and lower Cohen’s kappa for variables with smaller samples. The remaining images were then coded according to the established framework by one coder. Data was entered into a purpose-built database developed using REDCap electronic data capture tools hosted at The University of Sydney [22].

### 2.3. Data Analyses

Statistical analyses were performed using IBM SPSS Statistics, version 26.0 (Chicago, IL, USA). Descriptive statistics were conducted to summarize the distribution of coded categories for ‘Recent’ and ‘Top’ images. Chi-Square tests were performed to estimate whether ‘Recent’ images were different compared to ‘Top’ images. Post hoc Chi-Square Bonferroni adjustments were made to correct for Type I error where applicable. 

## 3. Results

At the time of capture there were a total of 1.3 million posts on Instagram linked to #postpartumbody. Of the 600 coded images included in the study, 29 were duplicate images (4.8%): 16 of which were posted in both the ‘Recent’ and ‘Top’ category, two appeared in the ‘Top’ category twice each, and 11 appeared in the ‘Recent’ category twice each. Overall, 126 (21%) images were part of a multiple series or video and 59 (9.8%) images were linked to a product or program. 

### 3.1. Content of Images

Table 2 outlines the categorisation of images. Overall, images categorised as food were mostly of nutrient-dense foods (n = 14, 74%) with only four images classified as energy-dense, nutrient-poor foods and one unable to be classified. Overall, 50 (8%) images were categorised as text-focused. Of these, 5 were not in English and 4 were unrelated to #postpartumbody despite being tagged with this hashtag. Most commonly, text-focused images included information about a specific diet/program/product (n = 14), motivational/positive text (n = 13) or information about a specific postpartum related product (n = 9). There were few exercise (n = 4) and nutrition information (n = 1) related text-focused images, and no images contained negative messaging or stigmatising text. There were 108 images of ‘Man only’, ‘Child only’, ‘Other group’ or ‘Pregnant woman’ representing 18% of images and 14 (2%) were coded as ‘Other’.

Of the 600 coded, 409 (68%) images were focused on a woman, including 256 (43%) coded as being of an individual woman, 59 (10%) coded as a before-and-after image, and 94 (16%) coded as being of a woman and a baby or child/ren. These ‘woman-focused’ images were further categorised according to adiposity, muscularity and pose (Table 3). Of the 409 images of women, 306 were able to be determined in respect to level of adiposity; 112 (37%) of these were of women coded with ‘low’ adiposity, 165 (54%) were of women coded with ‘average’ adiposity, and 29 (9%) were of women coded with ‘high’ adiposity. In total, 250 images were able to be determined in relation to muscularity, of which 130 (52%) were of a woman coded with ‘little to none’ muscularity, 110 images (44%) were of women coded with ‘visible definition’ and 10 images (4%) were of women coded with a ‘high level of definition’. The majority of the before-and-after images were of women who had experienced a reduction in weight (54%). Others were before-and-after images of pregnancy-to-postpartum (29%) or were body-building focused (12%).

Overall, poses in the 409 images of women were coded as being non-specific in 213 images (52%), in a sexual manner in 41 images (10%), fitness-focused in 88 images (22%), posing to accentuate a postpartum body feature in 21 images (5%) and a nurturing pose in 46 images (11%). Forty images displaying a nurturing pose were in the ‘woman and baby/child’ category, representing 44% of images in this category. The remaining six images of a nurturing pose were of women in before-and-after photos that also had a baby or child/ren in the image. The majority of women in woman-focused images were wearing either fitness/exercise attire (40%) or casual clothing (40%), followed by 8% in underwear and 5% in swimwear (Table 4). 

### 3.2. Recent Versus Top Images

Compared with ‘Top’ images, ‘Recent’ images were significantly more likely to be linked to a product or program (n = 49 (16%) in ‘Recent’, n = 10 (3%) in ‘Top’; *p* < 0.001) and more likely to be part of a multiple series or video (n = 93 (31%) in ‘Recent’, n = 33 (11%) in ‘Top’; *p* < 0.001). The categorisation of images also differed significantly between ‘Recent’ and ‘Top’ posts, χ2 = 65.38, 9 df, *p* < 0.001. Post hoc comparisons of categorisation of ‘Recent’ and ‘Top’ posts revealed that compared with ‘Top’ posts, ‘Recent’ posts were more likely to be categorised as text-focused or photos of food, and ‘Top’ images were more likely to be of a woman and her child/ren compared with ‘Recent’ images. The number of posts in other categories were similar in ‘Recent’ and ‘Top’ posts.

## 4. Discussion

This is the first study to describe the content of images on Instagram related to the hashtag #postpartumbody. Primarily, we found that the majority of images posted to this hashtag were woman-focused. Women in images generally had low or average levels of adiposity, little-to-none or some visible muscle definition and more than half were posing in a non-specific manner. Secondarily, compared with ‘Recent’ posts, ‘Top’ posts were less likely to be text-focused, less likely to be photos of food and less likely to be linked to a product or program, suggesting that, for this hashtag and population, Instagram posts that contain information are less ‘popular’ when compared with posts displaying images of people.

Despite the hashtag being #postpartumbody only 5% of images focused on features commonly associated with a postpartum body [23], including stretch marks, supple stomach, cellulite, sagging breasts or caesarean scars. Furthermore, our analysis identified that, of images able to be determined, only 9% were coded as being a woman with a ‘high’ level of adiposity with 37% coded as having ‘low’ and 54% coded as having ‘average’ adiposity. Given that the prevalence of women of reproductive age with a high body weight in increasing worldwide [24] and data reporting that many women gain more weight than recommended during pregnancy [25], our findings suggest that images viewed on Instagram related to #postpartumbody are not representative of the actual population of postpartum women. These findings are in line with objectification theory [11] and suggest that women with lower adiposity may be more likely to post images to Instagram tagged with #postpartumbody.

Although the majority of women in images included in this analysis were posing in a non-specific manner, approximately one third of women in images were posing in an objectifying manner, i.e., fitness-focussed or sexual manner. Similarly, a 2018 study examining the content of Instagram images tagged with #fitspiration found that many images contained objectifying elements [16]. In the age of the smart phone when multiple photos can be taken and edited, even photos of women with a non-specific pose are likely to have been carefully selected and possibly digitally enhanced before being posted to Instagram [8]. Women who view and compare themselves to such idealized imagery may be more likely to experience feelings of inadequacy and body dissatisfaction, particularly postpartum women who have recently undergone dramatic body changes during and following pregnancy [16].

In line with the body positive movement, whereby the narrow ideals of beauty are challenged, appearance-based social comparison are discouraged and bodies of any shape, size, or appearance are celebrated [23], it has been suggested that exposure to body positive content and a diverse array of women’s bodies on social media may improve body satisfaction [26,27]. An example of this includes a content analysis of Instagram images tagged with #curvyyoga which identified many images of people with a larger body size suggesting that, for this hashtag, Instagram was being used to promote yoga as a type of physical activity suitable for everybody [17]. In our study, the underrepresentation of images of women with ‘high’ adiposity, or women accentuating features characteristic of a postpartum body, suggests that, not only are images not representative of the population, but also that the body positive movement does not yet have a presence on social media in relation to the postpartum period.

The findings from our study support findings from a 2012 media content analysis of Australian magazine stories [28]. Both studies find evidence of publicly accessible media representing pregnancy as a state that needs to be ‘recovered’ from, similar to an illness. In our study, many women in images, even those not exhibiting a fitness-focused pose, are wearing fitness/exercise attire or doing some form of exercise. The frequency of these types of images suggests that women want to be seen to be exercising as a means of breaking the ‘hold’ that pregnancy had on them or ‘repairing’ their postpartum body. In the 2012 study, magazine stories about celebrity postnatal bodies identified social messages focused on expectations of the postpartum body to rapidly ‘bounce-back’ to the pre-pregnant state. Similarly, nearly half (46%) of the before-and-after images coded as part of this analysis were of a woman classified as having a ‘low’ level of adiposity, indicating rapid postpartum weight loss or little pregnancy weight gain. Furthermore, there may be more posting by women in this ‘low’ level adiposity category secondary to feeling more body pride, further supporting the objectification of a smaller and thinner body size and shape.

There has only been one previous study investigating the content of Instagram images that target women in the postnatal period. In this study, analysis of images and corresponding comments posted with popular breastfeeding-related hashtags (#breastfeeding, #breastmilk, #breastisbest, and #normalizebreastfeeding) found that Instagram was being used by mothers to support each other by sharing experiences and challenges in an overwhelmingly positive manner [29]. While it has been suggested that the visual imagery of an Instagram post contributes more to body image than any accompanying text [26], further research investigating the comments associated with images posted tagged with #postpartumbody may allow greater understanding of the response to images observed. Further, interviewing or surveying postpartum women may elucidate reasons for Instagram engagement and reactions to image content.

Other than images of women, the next most common category was text-focused images. Most of these images contained either positive/motivational messaging, information about a specific diet program or information about a specific postpartum related product. We found that 47 of the 51 text-focused images were a ‘Recent’ post rather than a ‘Top’ post indicating that text-focused images were less likely to be ‘popular’ or ‘trending’ compared with images of women. We also found that ‘Recent’ images were more likely to be linked to a product or program than a ‘Top’ post. Together, these findings suggest that women in the postpartum period may not be interested in information posted to Instagram, and this may include text-focused health related information. However, given that women are frequent users of Instagram, posting health information on Instagram may act to interrupt the idealized imagery observed in this study and prevent potential harm caused by such content.

Social media is highly accessed by women of reproductive age [3], however, more research is needed to determine why women use social media platforms such as Instagram, and whether social media is a suitable or preferred avenue for targeting public health messages around postpartum health including diet, exercise and psychological wellbeing. Indeed, the COVID-19 pandemic has demonstrated that postpartum women may be more likely than ever to access social media [30]. However, despite this increased use, they may not wish to obtain health information on this platform with research since the COVID-19 pandemic onset indicating that people prefer to receive health information from people they know and respect, including family, friends and community leaders such as religious leaders, rather than via social media, mainstream media, or healthcare professionals [31]. This highlights the importance of the connection people feel to the deliverer of health information, and should be considered in future health initiatives conducted via social media. Furthermore, it has been suggested that women engage more with their healthcare provider after searching on social media [32]. Together, these findings support further investigation into the potential utility of Instagram to convey important health information during the postpartum period.

Increasing use of various social media platforms highlights the need for this type of research to investigate, not only the impacts on the user, but also potential avenues for targeting future health related messaging and support. We were also able to base our methodology and coding framework on previously published papers [14,15,16,17,18].

One study limitation is data collection bias, whereby image sampling occurred at one timepoint only. This limitation could be overcome by repeat investigation. We did not investigate the content of the comments related to images. This could be an avenue for future research to determine reactions (i.e., positive or negative) to various types of images. Instagram posts shared by influencers or other users with large followings are more likely to be viewed and/or shared. Therefore, a post may appear in the ‘Top’ category as a result of the user’s popularity rather than image content. This was not accounted for in this study. Future research may overcome this limitation by employing an analytic approach to adjust for confounding factors. Coding in this type of study is subject to a degree of subjectivity. However, the coding framework, coder training and co-coding were conducted to increase the reliability of the data. Furthermore, while coding was conducted by two female coders of childbearing age, both coders are dietitians with PhDs and so their interpretation of images may be different to the general postpartum woman scrolling on Instagram. Additionally, little is known about how the user profile may influence the content displayed. For example, it is unknown how viewed images may change for postpartum women according to specific age, ethnicity, other interests on social media, etc. We created a new Instagram account for a female of average childbearing age in an attempt to be as specific to the population of interest as possible. Indeed, some viewers of images tagged with #postpartumbody may not be a woman in the postpartum period, but could be anyone given that images are in the public domain. As it was our intention to focus on images targeted at postpartum women, we have focused the discussion of our results in this context. Comparing ‘Recent’ and ‘Top’ images is limited as it is possible that images appear in both categories. In fact, we identified 16 images that appeared in both ‘Recent’ and ‘Top’ categories. Furthermore, ‘Recent’ images may indeed become ‘Top’ images in time. This could not be accounted for in our analysis. Videos were coded based only on the thumbnail image and, when applicable, only the first image of a series of multiple images was included.

## 5. Conclusions

The majority of the images on Instagram tagged with #postpartumbody are images of women: of determinable images 37% were of ‘low’ adiposity, 9% of ‘high’ adiposity and 53% in fitness attire. These findings suggest that women of lower adiposity, and in fitness attire, are more likely to post images of themselves on Instagram than women of higher adiposity. In line with objectification theory, viewing such imagery may worsen body satisfaction at this already vulnerable life stage. However, experimental research is required to determine this. Given that Instagram is highly accessed by women during the postpartum period, the inclusion of health information may be necessary to interrupt the potentially harmful content observed in our study.

## Figures and Tables

**Table 1 healthcare-10-01802-t001:** Codebook for *n* = 600 images uploaded to Instagram tagged with *#postpartumbody* including inter-coder reliability.

Category	Coded Items	% Agreement	Kappa
Type of post	Recent Top	100%	1.000
Video or multiple images?	Yes No	98.3%	0.956
Linked to product or program	Yes No	97.5%	0.787
Duplicate of a previous image?	Yes No	100%	1.000
Overall image category	Food (real food, not supplements) Pregnant woman Woman only Man only Child only Woman and baby/ies or child/ren Other group Before-and-after woman Text-focused Other	94.2%	0.924
Food	Nutrient-dense (i.e., fruits and vegetables, home-cooked meals, smoothie, portion plate type meals, foods as per dietary guidelines); Energy-dense (i.e., fast food, desserts, soft drink, alcohol, large portions); Unable to classify (e.g., mixed meal)	66.7%	0.400
Woman only	Face or body	97.7%	0.876
	Adiposity (low, slight frame with little to no visible fat stores; average, medium frame with moderate level of visible fat; high, high level of excess fat; unable to be determined due to the framing of the image or clothing covering the body)	68.2%	0.554
	Muscularity (little-to-none; visible definition; high-level definition; unable to be determined)	72.7%	0.628
	Pose (fitness/muscle focus, e.g., flexing muscle, or bodypart enhancing pose; posing in sexual manner, e.g., alluring/sultry gaze, winking or arching back, clevage showing; nuturing focus; posing to accentuate postpartum body feature, e.g., stretch marks or abdominal adiposity; non specific)	84.1%	0.740
	Attire (Casual; Fitness/exercise attire; Underwear; Not able to be determined; Swimwear; Formal)	82.2%	0.737
Woman and baby/ies or child/ren (coding woman only)	Face or body	100%	1.000
	Adiposity (low; average; high)	66.7%	0.528
	Muscularity (little-to-none; visible definition; high-level definition; unable to be determined)	83.3%	0.758
	Pose (fitness/muscle focus; sexual manner; nurturing focus; accentuating postpartum body feature; non specific)	83.3%	0.721
	Attire (Casual; Fitness/exercise attire; Underwear; Not able to be determined; Swimwear; Formal)	66.7%	0.447
Before and after of woman (with or without baby/child) (coding after image of woman only)	Type (Pregnancy only; Pregnancy to postpartum; Body building; Weight loss; other)	84.2%	0.687
	Adiposity (low; average; high)	84.2%	0.816
	Muscularity (little-to-none; visible definition; high-level definition; unable to be determined)	73.7%	0.615
	Pose (fitness/muscle focus; sexual manner; nurturing focus; accentuating postpartum body feature; non specific)	84.2%	0.752
	Attire (Casual; Fitness/exercise attire; Underwear; Not able to be determined; Swimwear; Formal)	85.0%	0.758
Text-focused	Not related to hashtag OR not in English	100%	1.000
	If related to hashtag: Positive text (e.g., motivational, inspirational) Negative text (e.g., stigmatising, unhelpful, inciting shame/guilt) Nutrition information (e.g., recipe, food swaps, facts) Specific diet/program/supplement Exercise related information (e.g., program, equipment) Behaviour change related (e.g., mindfulness, sleep) Postpartum product (e.g., cream for stretch marks, oils, baby advice/courses)	33.3%	0.333

**Table 2 healthcare-10-01802-t002:** Category of recent and top images linked to *#postpartumbody*, number (%).

Image Category	Recent (n = 300)	Top (n = 300)	Total (n = 600)
Woman only	118 (39)	138 (46)	256 (43)
Woman and baby/ies or child	**30 (10)**	**64 (21)**	94 (16)
Before and after woman (with or without baby/other in photo)	31 (10)	28 (9)	59 (10)
Text-focused	**46 (15)**	**4 (1)**	50 (9)
Group eg family, friends, two women	19 (6)	22 (7)	41 (7)
Man only	15 (5)	22 (7)	37 (6)
Food (real food, not supplements)	**17 (6)**	**2 (1)**	19 (3)
Child only	7 (2)	8 (3)	15 (3)
Pregnant woman	7 (2)	8 (3)	15 (3)
Other	10 (3)	4 (1)	14 (2)

Bold indicates significant difference (*p* < 0.05) between groups.

**Table 3 healthcare-10-01802-t003:** Categorisation of adiposity, muscularity and pose of ‘Recent’ and ‘Top’ images, number (%).

	Individual Woman			Before/After Woman			Woman and Baby/Ies/Child/Ren			All Women-Focused Images Combined		
	Recent (n = 118)	Top (n = 138)	Total (n = 256)	Recent (n = 31)	Top (n = 28)	Total (n = 59)	Recent (n = 30)	Top (n = 64)	Total (n = 94)	Recent (n = 179)	Top (n = 230)	Total (n = 409)
Head only	21 (18)	20 (14)	41 (16)	0 (0)	3 (11)	3 (5)	5 (17)	12 (19)	17 (18)	26 (15)	35 (15)	61 (15)
Type												
- Pregnancy to postpartum				5 (16)	12 (43)	17 (29)						
- Body building				3 (10)	4 (14)	7 (12)						
- Weight loss				22 (71)	10 (36)	32 (54)						
- Pregnancy only				0 (0)	1 (4)	1 (2)						
- Other				1 (3)	1 (4)	2 (3)						
Adiposity												
- Low	32 (27)	44 (32)	76 (30)	14 (45)	13 (46)	27 (46)	5 (17)	4 (6)	9 (10)	51 (28)	61 (27)	112 (27)
- Average	46 (39)	52 (38)	98 (38)	14 (45)	10 (36)	24 (41)	12 (40)	31 (48)	43 (46)	72 (40)	93 (40)	165 (40)
- High	10 (8)	7 (5)	17 (7)	3 (10)	1 (4)	4 (7)	4 (13)	4 (6)	8 (9)	17 (9)	12 (5)	29 (7)
- Not able to be determined	11 (9)	14 (10)	25 (10)	0 (0)	0 (0)	0 (0)	4 (13)	14 (22)	18 (19)	15 (8)	28 (12)	43 (11)
- Not applicable (head only)	19 (16)	21 (15)	40 (16)	0 (0)	3 (11)	3 (5)	5 (17)	11 (17)	16 (17)	24 (13)	35 (15)	59 (14)
- Not applicable (pregnant)	-	-	-	0 (0)	1 (4)	1 (2)	-	-	-	0 (0)	1 (0)	1 (0)
Muscularity												
- Little to none	36 (31)	44 (32)	80 (31)	12 (39)	10 (36)	22 (37)	9 (30)	19 (30)	28 (30)	57 (32)	73 (32)	130 (32)
- Visible definition	37 (31)	40 (29)	77 (30)	13 (42)	9 (32)	22 (37)	5 (17)	6 (9)	11 (12)	55 (31)	55 (24)	110 (27)
- High level definition	2 (2)	2 (1)	4 (2)	2 (7)	4 (14)	6 (10)	0 (0)	0 (0)	0 (0)	4 (2)	6 (3)	10 (2)
- Not able to be determined	24 (20)	31 (22)	55 (21)	4 (13)	1 (4)	5 (9)	11 (37)	28 (44)	39 (41)	39 (22)	60 (26)	99 (24)
- Not applicable (head only)	19 (16)	21 (15)	40 (16)	0 (0)	3 (11)	3 (5)	5 (17)	11 (17)	16 (17)	24 (13)	35 (15)	59 (14)
- Not applicable (pregnant)	-	-	-	0 (0)	1 (4)	1 (2)	-	-	-	0 (0)	1 (0)	1 (0)
Pose												
- Sexual manner	17 (14)	19 (14)	36 (14)	0 (0)	2 (7)	2 (3)	2 (7)	1 (2)	3 (3)	19 (11)	22 (10)	41 (10)
- Fitness-focused	37 (31)	29 (21)	66 (26)	8 (26)	8 (29)	16 (27)	4 (13)	2 (3)	6 (6)	49 (27)	39 (17)	88 (22)
- Postpartum body feature	6 (5)	5 (4)	11 (4)	3 (10)	0 (0)	3 (5)	1 (3)	6 (9)	7 (7)	10 (6)	11 (5)	21 (5)
- Nurturing	-	-	-	1 (3)	5 (18)	6 (10)	13 (43)	27 (42)	40 (43)	14 (8)	32 (14)	46 (11)
- Non-specific	58 (49)	85 (62)	143 (56)	19 (61)	13 (46)	32 (54)	10 (33)	28 (44)	38 (40)	87 (49)	126 (55)	213 (52)

**Table 4 healthcare-10-01802-t004:** Clothing worn by women in women-focused images (n = 409) posted to Instagram tagged with #postpartumbody, number (%).

Attire	Recent (n = 179)	Top (n = 230)	Total (n = 409)
Casual	44 (25)	119 (52)	163 (40)
Fitness/exercise attire	89 (50)	73 (32)	162 (40)
Underwear	16 (9)	16 (7)	32 (8)
Not able to be determined	13 (7)	10 (4)	23 (6)
Swimwear	12 (7)	8 (3)	20 (5)
Formal	5 (3)	4 (2)	9 (2)

## Data Availability

The data presented in this study are available on request from the corresponding author.

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
