# Peer review of "Content Analysis of #Postpartumbody Images Posted to Instagram"

_healthcare, 2022, doi:10.3390/healthcare10091802_

Round 1

Reviewer 1 Report

Justify the analysis in the abstract.

Improve motivation in the introduction.

Add a section containing the theoretical framework towards application of social media in digital health.

Discuss the results based on the theoretical section. 

The final section should be restructured. A systematic discussion of the findings is needed. 

In the conclusion section,  

a.       Provide a summary of the findings 

b.       Please highlight why your findings are so important 

c.       In what way, do the findings contribute to the current knowledge? 

d.       Answer to “so what” question 

Reviewer 2 Report

Summary:

This manuscript describes the results of a hashtag (#) analysis on Instagram. The use of a single hashtag (#postpartumbody) was examined during a single Sunday afternoon to Monday morning period in October 2020. A fake Instagram account for an imaginary Australian female of childbearing age (31.2 years) was used to access Instagram and gather the data. Two of the authors manually examined the search results from this time period, scoring the Instagram posts into ‘Top’ and ‘Recent’ categories as well as evaluating body characteristics of the images in the collected posts. The authors evaluate the data and make conclusions about conveying health-related information on Instagram.

Abstract:

The first line of the abstract about Instagram use being related to mood and body image satisfaction in young women is misleading given the design of this experiment. This statement is not relevant to the reported data as no information about mood and/or body image in women of any age are provided in the manuscript.

The second sentence of the Abstract inappropriately associates the number of hashtag uses with Instagram user ‘engagement’. There is no evidence that the number of hashtag uses reflects any sort of end user active ‘engagement’ – perhaps, the authors use of ‘engagement’ needs to be operationally defined better in this manuscript. It might be helpful if the authors separated the number of hashtag ‘likes’ ‘reposts’, or other forms of social messaging acknowledgement throughout the manuscript.

The second to last sentence in the Abstract should be removed or modified to reflect the fact that no data were collected in this study about the ‘actual’ body phenotype of women (the Instagram users) in the post-partum period, and an ‘idealized’ body type is highly culturally and socially determined. Furthermore, the manuscript contains no data about body dissatisfaction resulting from Instagram content was collected in this study – potential extensions of the presented results into this realm are an overreach. In other words, the lack of end user data makes any attempts to imply a response to the collected data nebulous.

1.       Introduction:

This reviewer takes issue with the approach of the authors stating that social media platforms are a primary source of information and behavior change in regards to healthcare issues, of which post-partum behavior is part. To support the authors supposition, one would need data comparing how postpartum behaviors are influenced by social media in comparison to family and traditional healthcare providers (physicians, nurses, etc.).

The authors seem to imply only postpartum women view images of other women in the post-partum period. Could fathers/men and/or currently pregnant women also be views of the Instagram images? If any of these, or other, groups view and derive information from these Instagram posts, much of what the authors are implying would not be pertinent.

2.       Materials and Methods:

There is no rationale provided for the selection of the sampling period selected. One would expect the number of hashtag uses of any topic to be influenced by celebrities and other popular figures who might have posted similar things on Instagram before/near the time examined. The generalizability of the results would be greatly enhanced if multiple time periods were collected and examined.

A rationale for the selection of #postpartumbody as opposed to other potential hashtags to evaluate the same sort of information is lacking. The authors state this hashtag is ‘highly utilised’ but no explanation of what this means is provided. Perhaps evaluating data collected using this hashtag in comparison to other related hashtags would be helpful?

There is no control for potential image manipulation in the dataset. Instagram is well-known for people using image manipulation software on posted images. The authors mention this possible confound in the Discussion, but ignore it in their experimental design and interpretation.

There is a possibility that images with before-after women might provide more healthcare or body image information after giving birth than images of women alone or women with children. A separate evaluation of these types of images might prove useful given the authors intent and propensity to focus on negative mental health of viewing post-partum images on Instagram. Perhaps data collection on the time between the before and after image would be meaningful?

The authors assume that the people (the audience) who look at images of women in the post-partum period are post-partum women – there is no evidence provided for this assumption.

3.       Results:

The Results presented are primarily descriptive (other than inter-evaluator kappa scores).

An explanation of the similarities and differences between ‘Recent’ and ‘Top’ images would be helpful.

4.       Discussion:

Just because this is the first study to describe the content of images on Instagram using hashtag postpartumbody, does not mean the study is well designed or the results well-evaluated in the context of the data collected. For example, this experiment suffers from a data collection bias. Only a single time period at a specific time of year was examined. This sampling limitation is a significant limitation of this study.

It is appropriate the authors note the observation that only 5% of images focused on stretchmarks, supple stomach, cellulite, sagging breasts, or Caesarean scars, suggest the posted images do not represent that actual population of women in the post-partum period. Instead, it suggests a certain type of person with a different focus posts afterbirth images on Instagram. The authors do note that this is evidence for a lack of a body-positive movement during the short period examined. Perhaps the data suggest ‘adiposity’ (or something else the authors measured) is a more meaningful reason people post postpartum pictures on Instagram? If this is the case, the authors miss an opportunity to make and elaborate on this point.

There are no data provided on the consequences of viewing Instagram images on viewer behavior. Extensions of the current data that suggest any sort of behavior change in post-partum women or others are conjecture. It is possible social media (e.g., Instagram) might have potential applicability to deliver healthcare messages to the public. However, if the data from the recent COVID-19 pandemic and vaccines are relevant, people do not respond well to ‘factual’ information delivered by healthcare representatives on social media or mainstream media. Instead, people tend to derive healthcare information primarily from friends, family, and religious leaders. This means it is not the ‘facts’ that are important for healthcare action, but the connection/believability of the deliverer. A connection to recent worldwide attempts to influence healthcare behavior would be valuable.

Interpretation of the data is too generalized for all women in the post-partum period. Who is the audience? Would the image of viewing images be different for women after her first child compared to her second, third, or more children? Would viewing images be different for mothers in their teens compared to 20’s, 30’s, or older? Does the educational level of the Instagram image viewer impact the response to the image? Indeed, would viewing images of women from same/different cultures have different impact factors on the viewing audience? Another way of looking at it is that there should be some consideration of what types of people post images, post-partum or otherwise, on Instagram? How might the characteristics of the end use influence susceptibility to what the authors describe?

5. Conclusions:

The statement ‘Women appear to be less engaged with information containing posts’ is representative of one main issue with this manuscript. No data are provided having anything to do with how anyone responds to viewing images in Instagram are provided or referenced.

Round 2

Reviewer 2 Report

The revised manuscript is improved. The concerns identified in this reviewers first comments are addressed. However, addressing these issues with how the manuscript is written do not solve the limitations associated with the experimental design. Most of the experimental design related limitations are now mentioned in the manuscript. Mentioning the shortcomings associated with the experiment reduces, but does not eliminate, the potential over-generalization of the data collected.